# Chemotherapy Drugs Act Differently in the Expression and Somatic Mobilization of the *mariner* Transposable Element in *Drosophila simulans*

**DOI:** 10.3390/genes13122374

**Published:** 2022-12-16

**Authors:** Taís Maus Bernardt, Estéfani Maria Treviso, Mariana Cancian, Monica de Medeiros Silva, João Batista Teixeira da Rocha, Elgion Lucio Silva Loreto

**Affiliations:** 1Biological Sciences, Federal University of Santa Maria (UFSM), Santa Maria 97105-000, RS, Brazil; 2Genetic and Molecular Biology, Federal University of Rio Grande do Sul (UFRGS), Porto Alegre 91501-970, RS, Brazil; 3Department of Biochemistry and Molecular Biology, Federal University of Santa Maria (UFSM), Av. Roraima 1000, Camobi, Santa Maria 97105-900, RS, Brazil

**Keywords:** *mariner*, cisplatin, dacarbazine, daunorubicin, *Drosophila*

## Abstract

Transposable elements (TEs) are abundant in genomes. Their mobilization can lead to genetic variability that is useful for evolution, but can also have deleterious biological effects. Somatic mobilization (SM) has been linked to degenerative diseases, such as Alzheimer’s disease and cancer. We used a *Drosophila simulans* strain, in which SM can be measured by counting red spots in the eyes, to investigate how chemotherapeutic agents affect expression and SM of the *mariner* TE. Flies were treated with Cisplatin, Dacarbazine, and Daunorubicin. After acute exposure, relative expression of *mariner* was quantified by RT-qPCR and oxidative stress was measured by biochemical assays. Exposure to 50 and 100 µg/mL Cisplatin increased *mariner* expression and ROS levels; catalase activity increased at 100 µg/mL. With chronic exposure, the number of spots also increased, indicating higher *mariner* SM. Dacarbazine (50 and 100 µg/mL) did not significantly alter *mariner* expression or mobilization or ROS levels, but decreased catalase activity (100 µg/mL). Daunorubicin (25 and 50 µM) increased *mariner* expression, but decreased *mariner* SM. ROS and catalase activity were also reduced. Our data suggest that stress factors may differentially affect the expression and SM of TEs. The increase in *mariner* transposase gene expression is necessary, but not sufficient for *mariner* SM.

## 1. Introduction

Transposable elements (TEs) are DNA sequences that can change their position in the genome [1]. They are ubiquitous, i.e., they occur in almost all living organisms; they are abundant, having long been an important component of several genomes, such as in *Drosophila* (20%) [2,3], maize (80%) [4,5], and humans (48%) [2,6]; and they are extremely variable in sequence, transposition mechanisms, and evolutionary history. Because of this, more than a hundred families are known [7,8]. TEs were originally considered “junk DNA”, but numerous studies now show that TEs drive evolution by offering new gene-splicing patterns and providing protein domains for coding genes that confer new cellular functions. Moreover, the presence of cis-regulatory sequences in TEs can rewire gene expression [9,10]. However, mobilization of TEs is generally deleterious, and several mechanisms have evolved to control transposition [11,12].

Somatic mobilization (SM), i.e., transposition of TEs in somatic tissues, was, until recently, a little-noticed phenomenon because it was considered rare and had no evolutionary or biological significance [13]. However, recent evidence has shown that SM is common, at least for some TEs, and may be associated with some age-related human diseases, such as Alzheimer’s disease and cancer [14,15]. For this reason, it is important to understand the SM process, and its regulation or deregulation. Stress is considered a regulator of the activation of TEs. Indeed, several studies have shown that TEs are activated by stress [5,16,17,18,19,20,21]. However, at the transcriptional level, this phenomenon is complex, as different stressors can elicit different responses in different TEs, with some being upregulated and others downregulated [22].

Chemotherapy drugs deserve special attention as potential TEs activators due to stress, as TEs are thought to play a role in generating genetic variability that induces resistance to cancer treatments. Cisplatin, daunorubicin, and dacarbazine are chemotherapy drugs largely used for cancer treatment. Cisplatin and dacarbazine are alkylating agents, with cisplatin binding DNA and promoting single-strand breaks, while dacarbazine inhibits DNA synthesis by acting as a purine analogue [23,24]. Daunorubicin has a different mechanism of action. It has a DNA-intercalating molecule that inhibits the enzyme topoisomerase II and stops the DNA replication process [25]. These drugs also promote oxidative stress [24,26].

*Drosophila* is one of the leading models for the study of TEs, as it has a well-annotated and -characterized genome and mobilome, in addition to an arsenal of research tools [27]. It is also important to note that 20% of *Drosophila*’s TEs are active [28]. Using the *Drosophila* model and a transcriptomic approach, we have previously shown that chemotherapy drugs, such as cisplatin and cyclophosphamide, cause changes in the expression of some TEs, activating some and suppressing others [29,30]. However, there are few studies investigating how chemotherapy can affect the expression and SM of TEs at the molecular and phenotypic levels. Here, we used the *Drosophila simulans white-peach* (*w^pch^*) strain to exploit its striking ability to somatically mobilize TEs. The *w^pch^* strain is a mutant strain with a non-autonomous *mariner-peach* in the *white* gene promoter, which produces peach-eyed flies due to the insertion of the *mariner*. This strain has the autonomous *mariner-Mos1* in the genome which produces a transposase responsible for excision of *mariner-peach.* This leads to mosaicism in the eyes, where some eye-forming cells revert to the red color of wild-type flies [31,32,33]. We tested whether the chemotherapy drugs cisplatin, dacarbazine, and daunorubicin alter the relative expression of *mariner* and the SM of *mariner*-*peach* in the *D. simulans w^pch^* strain. We also investigated whether these drugs cause oxidative stress upon acute exposure, and looked for the presence of regulatory regions for oxidative stress in the *mariner* sequence. Our data suggest that chemotherapy drugs may affect in expression and SM of TEs differently.

## 2. Materials and Methods

### 2.1. Fly Stocks Maintenance

Fly stocks were maintained at 20 °C and fed a medium of cornmeal, sucrose, and yeast. Female flies of the strain *D. simulans white-peach* (*w^pch^*), which were less variable in PCR assays [13], were used for all experiments. This strain has the autonomous TE *mariner-Mos1* in the genome and the non-autonomous *mariner-peach* TE copy in the *white* gene promoter, which produces peach-eyed flies due to the insertion of *mariner* [13].

### 2.2. Acute Exposure and Molecular Analysis

Young flies (1–4 days) were subjected to a 7-h starvation period [29] (Figure 1). Subsequently, the flies were transferred to culture media containing food coloring dye and one of the chemotherapy drugs (or only the dye, in the control treatment) for 24 h. After the exposure period, 20 females were selected for RNA extraction. All samples were taken in triplicate. Selection was done by visual analysis of the color of the abdomen (due to the presence of the red food dye) to standardize the samples. Thus, only flies that had ingested the culture medium and had similar abdomen color intensity were used.

The concentrations of each chemotherapy drug used were based on previously published studies. Cisplatin [34] and dacarbazine [35] were mixed to the culture media at 50 and 100 µg/mL, respectively. Daunorubicin was mixed to the media at a concentration of 25 and 50 µM [36].

RNA extraction was performed according to the TRIzol™ Reagent protocol (Invitrogen). RNA concentration was analyzed using the NanoDrop 2000 spectrophotometer (Thermo Fisher Scientific, Waltham, MA, USA). Samples were then treated with DNaseI (Promega, Madison, WI, USA) and synthesis of complementary DNA (cDNA) was performed using random primers according to the instructions of the High-Capacity cDNA Reverse Transcription Kit (Thermo Fisher Scientific).

The relative expression of the *mariner* transposase gene was measured through RT-qPCR on the StepOnePlus equipment (Applied Biosystems, Waltham, MA, USA) and SYBR Green was used as a fluorophore. The reaction was carried out in a final volume of 20 μL containing 10 μL of diluted cDNA (1:100), 0.25 U Platinum Taq DNA polymerase (Invitrogen, Waltham, MA, USA), 1× PCR reaction buffer, 3 mM MgCl_2_, 25 μM dNTPs 0.2 μM of each reverse and forward primer, and 1× SYBR Green (Molecular Probes, Eugene, OR, USA). To verify the relative expression of *mariner*, primers Mos308_F (GTGAACGGTGGTTTCAACG) and Mos490_R (AGCGATTGGAAACTGCTTGT) [13,37] were used, amplifying a 184 bp fragment. As endogenous control, the RP49 ribosomal protein gene was used, amplifying 121 bp with primers RP49_66-F (CCAGTCGGATCGATATGCTAA) and RP49_186-R (GTTCGATCCGTAACCGATGT) [38].

The RT-qPCR amplification was performed as follows: 95 °C for 7 min, followed by 40 cycles of 94 °C for 15 s, 60 °C for 15 s, and 72 °C for 20 s. Relative expression was calculated according to the 2^−ΔΔCT^ method [39]. Statistical analysis was performed by one-way Analysis of Variance (ANOVA), followed by Tukey’s multiple comparison test. Differences were considered significant when *p* < 0.05.

### 2.3. Chronic Exposure and Phenotypic Analysis

To determine the *mariner-peach* somatic mobilization rate (SM), chronic exposure to the drugs was performed at the lowest concentration used for acute exposure: 50 µg/mL cisplatin and dacarbazine, and 25 µM daunorubicin. For oviposition, adult flies were transferred to culture media containing one of the chemotherapy drugs (Figure 1). After 24 h, the adult flies were removed and the eggs were kept in the vials containing the respective drug throughout their life cycle. The same procedure was followed for the control group, but no drug was added to the medium. For phenotypic analysis, 300 female flies were collected for each group and visually examined for the presence or absence of red spots in the eyes. The collected data were compiled in a contingency table. For statistical analysis, Fisher’s exact test was performed and differences were considered significant when *p* < 0.05.

### 2.4. Biochemical Assays

The biochemical assays were adopted from Ogunsuyi and co-workers [40]. Acute exposure was performed as described in Section 2.2, except that no food coloring dye was added to the culture medium in order not to interfere with spectrophotometric measurements. After 24 h of exposure to one of the drugs, 10 female flies were selected from each vial and macerated in 0.1 M potassium phosphate buffer, and pH 7.4. Samples were centrifuged at 10.000 rpm for 10 min (Figure 1). The supernatant was recovered and the protein content was quantified using NanoDrop 2000. Samples were adjusted to a final concentration of 0.4 mg/mL of protein for all biochemical assays. All samples were taken in triplicate.

#### 2.4.1. DCFDA Assay

To determine the redox state of the cells, the 2′,7′-dichlorofluorescein diacetate (DCFDA) assay (Figure 1) was performed with a mixed solution of 180 μL of 0.1 M potassium phosphate buffer, pH 7.4, 2 μL of 1 M DCFDA and 16 μL of fly homogenate at 0.4 mg/mL, in a final volume of 198 μL. Fluorescence emission was monitored using a Spectramax spectrofluorometer (San Jose, CA, USA) at 30 s intervals for 30 min. The wavelengths used were 480 nm for excitation and 525 nm for emission. The statistical analysis was carried out by two-way ANOVA followed by post hoc Tukey’s test. Differences were considered significant when *p* < 0.05.

#### 2.4.2. Catalase Assay

Catalase activity was monitored by the consumption of H_2_O_2_ at 20 °C, at a wavelength of 240 nm, for 1 min. The reaction was performed in a 1 mL cuvette, containing 350 µL of 0.1 M potassium phosphate buffer, pH 7.4, 350 µL of 4 M H_2_O_2_, and 50 µL of fly homogenate at 0.4 mg/mL, in a final volume of 750 µL. The data were analyzed by two-way ANOVA followed by a post hoc test. Differences were considered significant when *p* < 0.05.

### 2.5. Search for Oxidative Stress Regulatory Regions

We searched for an Antioxidant Responsible Element (ARE) binding sequence in the regulatory region of *mariner* using RSAT http://rsat.sb-roscoff.fr/matrix-scan_form.cgi (accessed on 13 December 2022) [41]. The position-specific scoring matrix (PSSM) for the Nrf2-binding site in promoter was obtained in Malhotra and collaborators [42]. In addition, we manually searched for sequences similar to the ARE consensus sequence “TGACXXXGC” [42,43,44] in the 5′ UTR region of the *mariner* element.

## 3. Results

### 3.1. Cisplatin

Flies exposed to 50 and 100 µg/mL cisplatin showed a significant increase in expression at the transcriptional level of the *mariner* gene compared to the control group (*p* = 0.0036). There was no significant difference between the concentrations of 50 and 100 µg/mL (Figure 2A).

Since exposure to cisplatin increased mariner expression, we investigated whether the drug also altered *mariner-peach* SM, by performing phenotypic analysis to identify the presence of red spots in the eyes of the flies. We found that the group chronically exposed throughout their life cycle to 50 µg/mL cisplatin had more flies with spots than the control group (*p* < 0.0001, Figure 2B).

Catalase activity was significantly higher in protein extracts from flies treated with 100 µg/mL of cisplatin (*p* < 0.0001). On the other hand, there was no significant difference in catalase activity in the flies treated with 50 µg/mL cisplatin (Figure 2C). The overall ROS content in protein extracts of the flies exposed to both cisplatin concentrations increased significantly (*p* < 0.0001) compared to the non-treated group (Figure 2D). A dose dependence was observed within the two concentrations tested (*p* < 0.0001).

### 3.2. Dacarbazine

Acute exposure to 50 and 100 µg/mL dacarbazine did not significantly alter the relative expression levels of *mariner* in any comparison (*p* = 0.0712, Figure 3A). The levels of SM were also unaffected (*p* = 0.1176), as there was no significant difference in the number of flies with red eye spots after chronic exposure (Figure 3B).

There was a significant (*p* = 0.0133) decrease in catalase activity between the flies exposed to 100 µg/mL dacarbazine and the control group (Figure 3C). There was no difference in catalase activity or between the two concentrations in the flies treated with 50 µg/mL dacarbazine. Moreover, the flies treated with both concentrations of dacarbazine showed no significant change in fluorescence values for the DCFDA assay (*p* > 0.05) compared to the control group (Figure 3D).

### 3.3. Daunorubicin

Exposure to daunorubicin for 24 h significantly (*p* = 0.0004) increased the relative expression of the mariner gene at both concentrations compared with the control group (Figure 4A). In contrast, chronic exposure significantly decreased *mariner-peach* SM expression despite the increase in relative expression of mariner (*p* = 0.0232, Figure 4B).

Flies exposed to Daunorubicin presented a significant decrease (*p* < 0.05) in the overall ROS content intracellularly for both concentrations tested, as seen by the depletion of the fluorescence values (Figure 4D). Similarly, there was a decrease in catalase activity at the 50 µM exposure (*p* < 0.0001, Figure 4C).

### 3.4. ARE Binding Sequences

Analysis using RSAT server and the scoring matrix for Nrf2 described in [42] revealed only one binding site (*p* = 4.4 × 10^−5^). However, this site is located within the coding region and in the reverse strand and, based on these properties, should probably not be involved in transposase regulation. On the other hand, we found three ARE sites with some similarity to the consensus sequence of the Nrf2 site in the 5′UTR region of *mariner* (Figure 5C). We create a matrix with these sequences and reanalyze the *mariner* sequence using this matrix. One of these sites (ARE1) was indicated by RSAT as a putative binding site (*p* = 6.4 × 10 ^−5^). The promoter region of the *mariner* element with the putative ARE binding site is shown in Figure 5.

## 4. Discussion

Nrf2 (nuclear factor E2 p45-related factor 2) is a transcription factor (TF) that mediates one of the major defense mechanisms against oxidative stress. The fly homolog of Nrf2 is the Cap’n’collar (CncC) protein. This TF binds to the cis-regulatory sequence ARE and stimulates the expression of detoxification enzymes, such as catalase, glutathione S-transferase, peroxiredoxin, heme oxygenase-1, and others [42,43,44,45]. Nrf2 also promotes the expression of enzymes involved in the degradation of xenobiotics [43]. The chemotherapy drugs cisplatin, dacarbazine, and daunorubicin are xenobiotics known to induce oxidative stress. Therefore, the presence of a putative ARE sequence in the promoter region of the *mariner* transposase gene is expected to lead to activation of this gene in the presence of these agents (Figure 5B). It is worth noting that the position of the ARE sequence in the promoter region of the *mariner* transposase gene is similar to that of artificial promoters, which have been shown to be efficient in activating reporter genes under oxidative stress conditions [44]. It should be noted, however, that the ARE site that we suspect is present in *mariner* is only a guess and experimental tests are needed to verify if it is functional.

We had expected that increased activation of *mariner* would lead to an increase in flies with red spots in the eyes. However, this was not the case with all drugs. Cisplatin increased the general ROS level and catalase activity, indicating oxidative stress. As expected, an upregulation of the transposase gene and an increase in *mariner* SM was observed, as indicated by the number of flies with red spots in the eyes. For dacarbazine, no increased oxidative stress was detected, since there was no difference in the general ROS levels and catalase activity decreased for the highest concentration tested. There was also no change in *mariner* expression or SM in dacarbazine exposure. It can be suggested that the drug concentration used was below that capable of inducing oxidative stress in *Drosophila*, in contrast to mammals in which it was previously tested [35].

Daunorubicin, at the concentrations used, induced a decrease in catalase activity and ROS levels. There was also higher *mariner* expression, but lower SM. Doxorubicin, a drug that is similar to daunorubicin, also presented a decrease in ROS damage on breast cancer cell lineages [46]. However, increased levels of H_2_O_2_ and superoxide dismutase expression have been found [46]. We suggest that a significant decrease in ROS levels improved the cell’s ability to handle the drug and control the activity of TEs.

We found that the *mariner* transposase gene can be upregulated even if no oxidative stress was detected. The *mariner* element has other regulatory elements, such as HSE, which is activated by heat shock TF. However, this TF may be sensitive not only to heat [47]. The upregulation of *mariner* could be induced by regulatory elements other than ARE, or maybe even by ARE, but through the action of daunorubicin as a xenobiotic instead of oxidative stress. A remarkable observation is that an increase in transposase transcription was followed by a decrease in *mariner* mobilization, as measured by the number of flies with red eye spots. TE mobilization is a process involving several steps, and the transcription of genes encoding the enzymes for transposition is among the first ones. A possible explanation for the reduction in *mariner* mobilization could be the fact that daunorubicin is a DNA intercalator. Intercalating substances can distort the DNA helix and, in this way, reduce the binding of the enzyme to the DNA. An alternative hypothesis is that daunorubicin is a topoisomerase II inhibitor, and normally, transposition requires DNA replication [48]. Alterations caused by the drug in DNA replication may affect transposition. Another explanation could be an inhibitory effect of transposase over expression [49]. Therefore, it is possible that *mariner* transposase expression is greatly increased by daunorubicin upon chronic exposure, but mobilization does not occur and is still reduced because little or no transposase activity occurs.

The literature repeatedly emphasizes that stress activates the mobilization of TE, and this fact is supported by several lines of evidence. However, the abundance of transcriptomes recently obtained in organisms under different stress conditions has revealed a more complex scenario: a given stress activates one subset of TEs and represses others. Different stress factors promote different responses [22]. Similarly, the data shown here also suggest that the stressors used elicited different responses for the *mariner* transposon. The most important finding, however, is that even if a drug can increase transcription of transposon genes, other steps of transposition may be affected and the end result may be a reduction in mobilization.

## 5. Conclusions and Perspectives

In summary, different chemotherapeutic agents may have different effects on the same TE. Moreover, transposase gene expression is necessary but not sufficient for TE mobilization. These results contribute to understanding how TEs respond to different stresses and that different methods beyond mere expression are required to conclude that a TE is indeed mobilized. Questions remain about TEs activated by stress: is there a pattern to the activation of TEs? Which TEs are active during which stress? Does stress result in different responses at different developmental stages? Can the accumulation of different levels of stress alter the response of TEs? Future work could focus on exploiting this variability in stress response to target treatment of cancer and other diseases. Moreover, the choice of chemotherapeutic agent used in cancer treatment could take into account the activity of TEs to reduce side effects and/or improve treatment efficacy, as appropriate.

## Figures and Tables

**Figure 1 genes-13-02374-f001:**
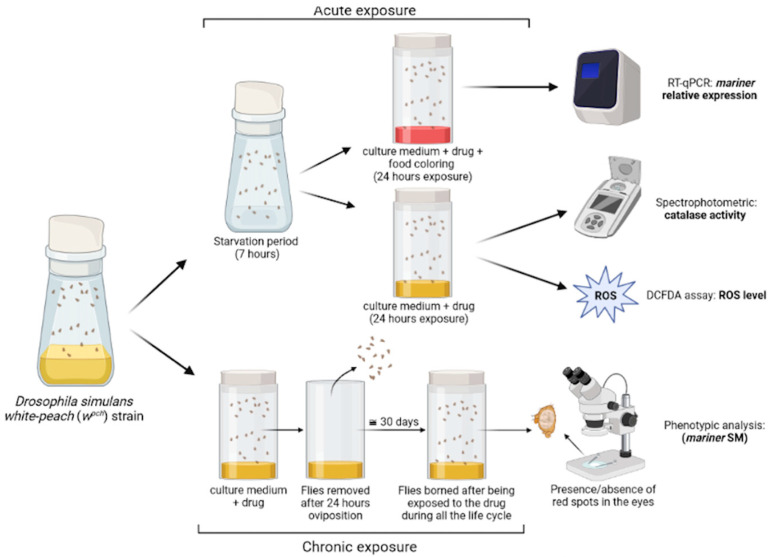
Experimental design. Acute exposure: 1-to-4-day-old *D. simulans w^pch^* flies were subjected to a 7-h starvation and then transferred to culture media containing cisplatin, dacarbazine, or daunorubicin plus food coloring dye for further analysis. No dye was added to the culture media of the flies used for the biochemical analyses. Chronic exposure: flies were oviposited for 24 h in culture media containing one of the active compounds, and the new generation was collected and analyzed. Created with BioRender.com, accessed on 25 October 2022, Licence AD24KGDIHQ.

**Figure 2 genes-13-02374-f002:**
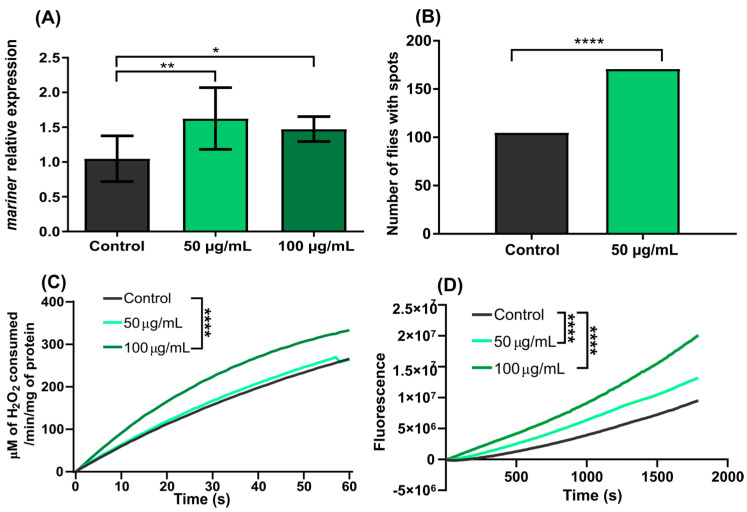
Effects of cisplatin in *D. simulans w^pch^*. (**A**) Relative transcriptional expression of the mariner TE in flies exposed to 50 and 100 µg/mL cisplatin. (**B**) Number of flies with red spots in the eyes after chronic exposure to 50 µg/mL cisplatin. (**C**) Monitoring of catalase activity in fly protein extracts, after 24 h of exposure to 50 or 100 µg/mL cisplatin. (**D**) Monitoring of ROS content in fly protein extracts, after 24 h of exposure to 50 or 100 µg/mL cisplatin. Data were analyzed by two-way ANOVA, followed by Tukey’s post hoc test. * Mean values are significantly different compared to Control (* = *p* < 0.05; ** = *p* < 0.001; **** = *p* < 0.0001).

**Figure 3 genes-13-02374-f003:**
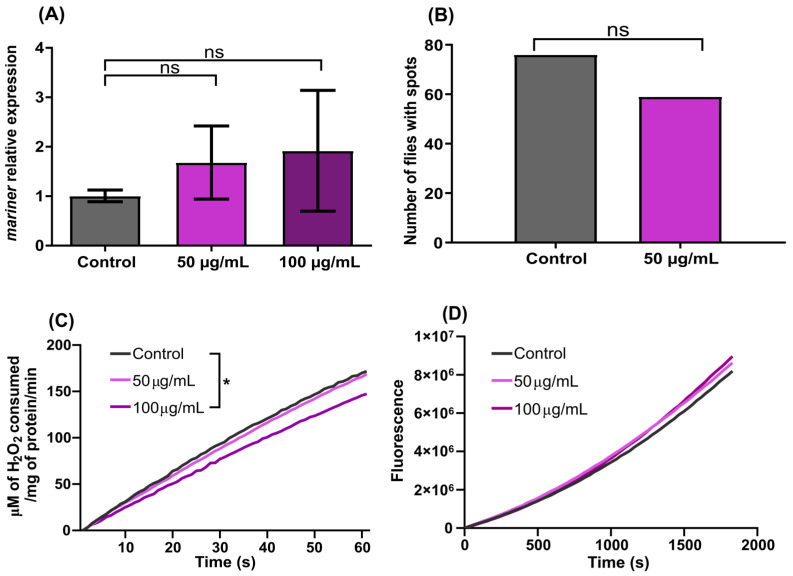
Effects of dacarbazine in *D. simulans w^pch^*. (**A**) Relative expression of the mariner TE in flies exposed to 50 and 100 µg/mL dacarbazine. (**B**) Number of flies with red spots in the eyes after chronic exposure to 50 µg/mL dacarbazine. (**C**) Monitoring of catalase activity after 24 h of exposure to 50 or 100 µg/mL dacarbazine. (**D**) Monitoring of ROS content in flies after 24 h of exposure to 50 or 100 µg/mL dacarbazine. Data were analyzed by two-way ANOVA, followed by Tukey’s post hoc test. * Mean values are significantly different compared to Control. Ns = non significant.

**Figure 4 genes-13-02374-f004:**
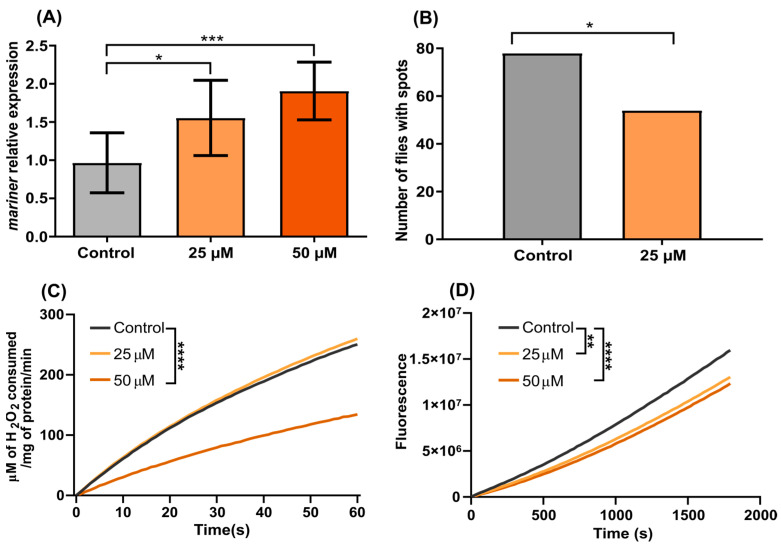
Effects of daunorubicin in *D. simulans w^pch^*. (**A**) Relative expression of the *mariner* gene in flies treated with 25 and 50 µM daunorubicin. (**B**) Number of flies with red spots in the eyes after chronic exposure to 25 µM daunorubicin. (**C**) Monitoring of catalase activity after 24 h of exposure to 25 and 50 µM Daunorubicin. (**D**) Monitoring of ROS content in flies after 24 h of exposure to 25 and 50 µM Daunorubicin. Data were analyzed by two-way ANOVA, followed by Tukey’s post hoc test. * Mean values are significantly different compared to Control (* = *p* < 0.05; ** = *p* < 0.01; *** = *p* < 0.001; **** = *p* < 0.0001).

**Figure 5 genes-13-02374-f005:**
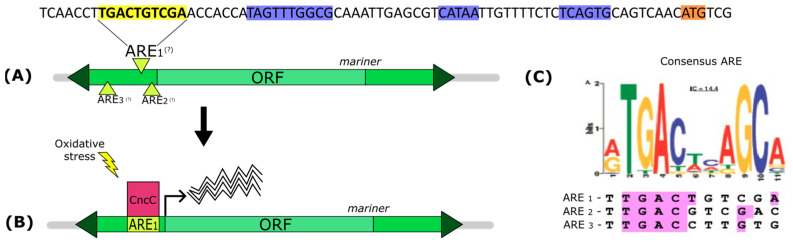
Promoter region of the *mariner* transposase gene. (**A**) The sequence highlighted in yellow is the putative binding site ARE (Antioxidant Responsible Element). The start codon is shown in orange and the core promoter elements are represented in blue. (**B**) The transcription factor CncC binds to the putative ARE binding sequence of *mariner* when exposed to oxidative stress, which increases the expression of the *mariner* transposase gene. (**C**) The logo of the consensus sequence ARE [41] and the three similar sequences to “ARE” found in the 5′ UTR region of *mariner*. Only “ARE 1” was statistically significant (*p* = 6.4 × 10^−5^).

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
