# Peer review of "Chemotherapy Drugs Act Differently in the Expression and Somatic Mobilization of the mariner Transposable Element in Drosophila simulans"

_genes, 2022, doi:10.3390/genes13122374_

Round 1

Reviewer 1 Report

Overall, this manuscript by Bernardt et al. is quite interesting. The authors used D. simulans white-peach strain to exploit its ability to mobilize transposable elements somatically, to investigate the effect of chemotherapy drugs (Cisplatin, Dacarbazine, and Daunorubicin) on mariner transposase expression and somatic mobilization of TE. In this study, the authors conducted acute and chronic drug exposure experiments to quantify changes in mariner transposase expression, oxidative stress levels, and scoring for flies with red spots in the eyes indicating mariner SM. Although they reported interesting results, each chemotherapy drug produced different effects in the somatic mobilization of TE.

Text issue:

Line 36-39: add reference.

Line 236-240: This should be part of the discussion, not results.

Experimental issues:

Chronic exposure (Line 130): This is the most crucial section of the paper since chronic exposure is the main experiment for understanding drug effects on somatic mobilization. Authors need to clarify the following issues

-       How stable are drugs in fly food over the fly's entire life cycle?

Or refer to previous studies that have used these drugs for longer than 24 hours.

-       Line 137-139, 

Why was the number/area of red spots in the eyes not scored upon drug exposure? The authors stated in Line 16 "SM can be measured by counting red spots in the eyes." These results can significantly strengthen the manuscript.

-       Additional suggestion, mariner relative expression analysis can be performed on 3rd instar wandering larvae or adult flies from chronic exposure experiments to complement phenotypic analysis results.

Line 142: Was one set of ten flies used for biochemical assays for each drug exposure? If yes need more replicates.

Author Response

#Reviewer 1

Overall, this manuscript by Bernardt et al. is quite interesting. The authors used D. simulans white-peach strain to exploit its ability to mobilize transposable elements somatically, to investigate the effect of chemotherapy drugs (Cisplatin, Dacarbazine, and Daunorubicin) on mariner transposase expression and somatic mobilization of TE. In this study, the authors conducted acute and chronic drug exposure experiments to quantify changes in mariner transposase expression, oxidative stress levels, and scoring for flies with red spots in the eyes indicating mariner SM. Although they reported interesting results, each chemotherapy drug produced different effects in the somatic mobilization of TE.

Text issue:

Line 36-39: add reference.

#Answer: Altered in the text.

 Line 236-240: This should be part of the discussion, not results.

#Answer: Altered in the text.

 Experimental issues:

Chronic exposure (Line 130): This is the most crucial section of the paper since chronic exposure is the main experiment for understanding drug effects on somatic mobilization. Authors need to clarify the following issues

#QUESTION 1 - How stable are drugs in fly food over the fly's entire life cycle? Or refer to previous studies that have used these drugs for longer than 24 hours.

#Answer: We did not measure the stability of the drugs in the fly food; however, some other studies have already tested the use of chemotherapeutic agents for more than 24 hours. On cisplatin (Groen et al, 2018; Podratz et al, 2016) for 5 and 3 days, respectively; on cyclophosphamide (Stoffel et al, 2020) for the entire life cycle; on daunorubicin (Chakraborty et al, 2018) for 3 and 7 days, and (Lehmann et al, 2004) for three days after oviposition until the end of their development.

#QUESTION 2 - Line 137-139, Why was the number/area of red spots in the eyes not scored upon drug exposure? The authors stated in Line 16 "SM can be measured by counting red spots in the eyes." These results can significantly strengthen the manuscript.

#Answer: We tested counting the number of spots and measuring the area of red spots to quantify somatic mobilizationin previous work https://doi.org/10.1016/j.gene.2018.08.079. In that study, we concluded that the simpler and easier method to analyze SM is to observe the presence or absence of red spots. For this reason, we only examined the presence or absence of red spots in the present study.

#QUESTION 3 - Additional suggestion, mariner relative expression analysis can be performed on 3rdinstar wandering larvae or adult flies from chronic exposure experiments to complement phenotypic analysis results.

 #Answer: The mariner element is somatically active at all developmental stages and its activity changes during the larval period (https://doi.org/10.1016/j.gene.2018.08.079). Because chemotherapy drugs can affect the rate of development, it is difficult to compare the activity of treated and control larvae because they may not be at similar developmental stages. For this reason, we decided to compare adults of the same age after acute treatment.

#QUESTION 4 - Line 142: Was one set of ten flies used for biochemical assays for each drug exposure? If yes need more replicates.

#Answer: We used three sets of ten flies for biochemical tests for each treatment and control. We have modified and clarified the information in the text.

#QUESTION 5 - The title of the manuscript sounds more like conclusion. Suggest to revise and change the title if possible. Also, the study was conducted on a single species D. simulans, so should use species name instead of only the genus name Drosophila in title.

#Answer: Altered in the text.

Reviewer 2 Report

The manuscript “Chemotherapy drugs act differently in the expression and somatic mobilization of the mariner transposable element” by Bernardt et al., addresses an important and timely topic as chemotherapy drugs increases the SM and thus inadvertently contributes to the progression of severe diseases. In this context, the present study raises the concern to carry out more extensive studies regarding these and other related drugs and xenobiotics.  

Overall, the study seems well conducted and very well written. The results are well organized, and the methods are explained in reproducible manner. Specific comments are mentioned below.

1.      The study very well explains the varying effects of different chemotherapy drugs on the mobilization of mariner TE; however, the authors does not explain the significance of these outcomes in context to its application in future. The authors should add a brief section at the end of discussion explaining the future directions in order to avoid and/or minimize these adverse effects.

2.      As mentioned in the manuscript, the flies were fed with dye mixed food and the flies with only colored abdomen were selected for further analysis and thus the flies that didn’t ingest the food were eliminated, however, did the authors check the amount of drug ingested by the flies through any method. The amount of food ingested can be of varying degree and thus the consequent phenotypes can also vary.

3.      Introduction last paragraph- In addition to explaining what the authors set out to elucidate through the study, they should also in very brief, add what they observed (what the overall results were as mentioned in the abstract).

4.      What was the reason to only use females for the study. Specify the same in the manuscript.

5.      Why were the flies that were subjected to biochemical assay were not fed with the dye. What if the flies that were chosen for these analyses did not ingest the drug mixed food.

6.      Results section 2.1 Cisplatin- There was no significant difference in the mariner expression between the two cisplatin concentrations, is this the reason of only using one cisplatin concentration (50µgm/mL) to analyze TE mobilization through red spots in eyes, however both concentrations were used for ROS and catalase activity. Same for Dacarbazine and Daunorubicin.

7.      Figure 2C, 2D, 3C, 3D, 4C and 4D- there is no indication of significance in all of the mentioned graphs as compared to control. Please specify the same accordingly.

8.      Results section 3.2 Dacarbazine- Please explain Figure 3C and 3D in the result section in the same sequence as they appear in the figure panel (First 3C and then 3D).

Author Response

#Reviewer 2 

The manuscript “Chemotherapy drugs act differently in the expression and somatic mobilization of the mariner transposable element” by Bernardt et al., addresses an important and timely topic as chemotherapy drugs increases the SM and thus inadvertently contributes to the progression of severe diseases. In this context, the present study raises the concern to carry out more extensive studies regarding these and other related drugs and xenobiotics. Overall, the study seems well conducted and very well written. The results are well organized, and the methods are explained in reproducible manner. Specific comments are mentioned below.

#QUESTION 1- The study very well explains the varying effects of different chemotherapy drugs on the mobilization of mariner TE; however, the authors does not explain the significance of these outcomes in context to its application in future. The authors should add a brief section at the end of discussion explaining the future directions in order to avoid and/or minimize these adverse effects.

#Answer: Altered in the text.

#QUESTION 2 - As mentioned in the manuscript, the flies were fed with dye mixed food and the flies with only colored abdomen were selected for further analysis and thus the flies that didn’t ingest the food were eliminated, however, did the authors check the amount of drug ingested by the flies through any method. The amount of food ingested can be of varying degree and thus the consequent phenotypes can also vary.

#Answer: We recognize that it is important to standardize the experiments and to make sure that the flies ingest a certain amount of drug, an interesting point. However, we have not found a method to measure this amount given the size of the flies and the conditions of the experiment. We tried to minimize the variation by using colored food and selecting only the flies with colored abdomen for further analysis, but the amount of drug ingested is a limitation of this experiment. Also, we use three sets of each treatment and a substantial sample size in each set to reduce variability.

#QUESTION 3 - Introduction last paragraph- In addition to explaining what the authors set out to elucidate through the study, they should also in very brief, add what they observed (what the overall results were as mentioned in the abstract).

#Answer: altered in the text

#QUESTION 4 - What was the reason to only use females for the study. Specify the same in the manuscript.

#Answer: Considering that the mariner-peach element is located on chromosome X, we think it is important to standardize the use of females or males for the experiment. We chose to use females because lower variability has been shown in previous work (data not shown). This was also stated in the manuscript.

#QUESTION 5 - Why were the flies that were subjected to biochemical assay were not fed with the dye. What if the flies that were chosen for these analyses did not ingest the drug mixed food.

#Answer: We tried using dye in the biochemical experiments, but as we homogenize a fly in potassium phosphate and the dye is dark enough to see in the fly's abdomen, the homogenizate was pale pink. We use a spectrophotometer and think that this color may affect our results. Also, the flies stay in the 7- hours starvation and the 24-hour treatment. We select the flies with the largest abdomen, indicating that they have been feeding. This information was included in the text

#QUESTION 6 - Results section 2.1 Cisplatin- There was no significant difference in the mariner expression between the two cisplatin concentrations, is this the reason of only using one cisplatin concentration (50µgm/mL) to analyze TE mobilization through red spots in eyes, however both concentrations were used for ROS and catalase activity. Same for Dacarbazine and Daunorubicin.

#Answer - The experiments were divided in chronic exposure (all larval stage) and acute exposure (adult flies during 24 h). We used just the lower concentration in the chronic exposure because they stay a long period eating the drugs, while the acute exposure happening during 24 h. The mariner expression and biochemical assays are performed after acute exposure, so we use both concentrations, while SM, through red spots in eyes, is a chronic exposure and we use just the lower concentration.

#QUESTION 7 - Figure 2C, 2D, 3C, 3D, 4C and 4D- there is no indication of significance in all of the mentioned graphs as compared to control. Please specify the same accordingly.

#Answer: Altered in the text and figures.

#QUESTION 8 - Results section 3.2 Dacarbazine- Please explain Figure 3C and 3D in the result section in the same sequence as they appear in the figure panel (First 3C and then 3D).

#Answer: Altered in the text.

Reviewer 3 Report

The manuscript titled “Chemotherapy drugs act differently in the expression and somatic mobilization of the mariner transposable element” describes how three chemotherapy drugs (Cisplatin, Dacarbazine, and Daunorubicin) can alter the relative expression of mariner based on controlled experiments with D. simulans wpch strain. The study animals are fascinating organisms-model to study somatic transposition, the methodological approaches are correctly addressed to access biological questions proposed, and results are shown clearly. 

I hope the authors appreciate my suggestions, representing my best efforts to provide constructive feedback for improving the final manuscript.

1. The manuscript objectively showed the major biological problem that was evaluated and the importance of the choice of the organism model proposed to the development of this study. However, the introduction presents a little background about the biological-model D. simulans white-peach strain. Including one short paragraph with better characterization of genotype x phenotype and genetic mechanisms that involve the somatic mobilization of these non-autonomous elements can improve readers’ understanding of the elegance can be the experimental model employed in this study. 

2. Add information (lines 119-120) about which region of the element was used to check the expression of the mariner. If the set of primers has already been described previously, please cite the literature reference.

3. (Line 236) The search for oxidative stress regulatory regions needs to be better characterized in the methodology. How were the sequences obtained? Database based? Or isolated and sequenced in the present study?

4. In Figure 5, scheme “b” represent a suggestion of the mechanism that increases the expression of the mariner transposase gene or already have been experimental pieces of evidence to validate this hypothesis? If this mechanism has already been validated, please include references about it. Negatively, make clear that the scheme is an executive suggestion about this mechanism of this respective gene (see lines 259-260). In my opinion, the Author didn’t address experimental evidence in this work in n affirmative sentence in the legends of figure 5, and new and specific experimental studies should be performed to validate this proposal.   

5. The conclusion of this work, as described in the 302-306 line, needs to demonstrate what the fundamental contributions of this manuscript are and is redundant in repeating the general results of the work. The conclusion should explore the contributions of this manuscript in the context of response to stress conditions and somatic mobilization and address where are the biological questions that remain open or new questions that arise with its results. Please carefully consider these issues when rewriting the conclusions of this paper.

Author Response

#Rewier 3

The manuscript titled “Chemotherapy drugs act differently in the expression and somatic mobilization of the mariner transposable element” describes how three chemotherapy drugs (Cisplatin, Dacarbazine, and Daunorubicin) can alter the relative expression of mariner based on controlled experiments with D. simulans wpch strain. The study animals are fascinating organisms-model to study somatic transposition, the methodological approaches are correctly addressed to access biological questions proposed, and results are shown clearly.

I hope the authors appreciate my suggestions, representing my best efforts to provide constructive feedback for improving the final manuscript.

#QUESTION 1 - The manuscript objectively showed the major biological problem that was evaluated and the importance of the choice of the organism model proposed to the development of this study. However, the introduction presents a little background about the biological-model D. simulans white-peach strain. Including one short paragraph with better characterization of genotype x phenotype and genetic mechanisms that involve the somatic mobilization of these non-autonomous elements can improve readers’ understanding of the elegance can be the experimental model employed in this study.

#Answer: Altered in the text.

#QUESTION 2 - Add information (lines 119-120) about which region of the element was used to check the expression of the mariner. If the set of primers has already been described previously, please cite the literature reference.

#Answer: Altered in the text.

#QUESTION 3 - (Line 236) The search for oxidative stress regulatory regions needs to be better characterized in the methodology. How were the sequences obtained? Database based? Or isolated and sequenced in the present study?

#Answer:  We have performed new analyses with the RSAT software to clarify this part. These analyses are included in the new version.

#QUESTION 4 - In Figure 5, scheme “b” represent a suggestion of the mechanism that increases the expression of the mariner transposase gene or already have been experimental pieces of evidence to validate this hypothesis? If this mechanism has already been validated, please include references about it. Negatively, make clear that the scheme is an executive suggestion about this mechanism of this respective gene (see lines 259-260). In my opinion, the Author didn’t address experimental evidence in this work in n affirmative sentence in the legends of figure 5, and new and specific experimental studies should be performed to validate this proposal.  

#Answer:  Indeed, we have no experimental evidence that the suspected ARE binding sites are functional. The evidence was provided only by bioinformatics. We highlight this point in the text.

#QUESTION 5 - The conclusion of this work, as described in the 302-306 line, needs to demonstrate what the fundamental contributions of this manuscript are and is redundant in repeating the general results of the work. The conclusion should explore the contributions of this manuscript in the context of response to stress conditions and somatic mobilization and address where are the biological questions that remain open or new questions that arise with its results. Please carefully consider these issues when rewriting the conclusions of this paper.

#Answer: Altered in the text.

Reviewer 4 Report

This manuscript describe the mobilization of a mariner element inserted in the white gene of D. simulans, upon exposure to chemotherapy drugs, such as Cisplatin, Dacarbazine, and Daunorubicin.

In my opinion the manuscript is well written and the conclusions are supported by the results obtained. 

However, I have some major points that I think the authors should clarify. More important, I have found the manual annotation of the ARE elements not appropriate. This analysis should be repeated in a more rigorous way. Finally, some figures should be better presented.

Please, find my specific comments below.

L36-39 In the context of the functional role that TEs have in the eukaryotic genome, it would be appropriate to cite relevant reviews that have summarized this topic (e.g  10.3390/biology9020025 and 10.1098/rstb.2019.0347)

ARE detection. The authors state that the detection of ARE motifs was made manually. They should use more reliable methods such to perform detection, possibly associated to a reliability score or similar.

More important, I suggest to check carefully how this analysis was performed. I suspect there is an error in the reported ARE motif (TGACNNNCG) that should be RTGACNNNGC as reported in Itoh et al. 1999, and as also reported in the PWM described in the work of Malhotra et al (10.1093/nar/gkq212)

I suggest using tools in RSAT, such as the matrix scan tool (http://rsat.sb-roscoff.fr/matrix-scan_form.cgi), which detect motifs based on PWMs. Nrf2 PWM has been described in the work of Malhotra et al (10.1093/nar/gkq212).

How many ARE motifs can be detected? If many, are they clustered? 

Figure 5. This figure could be better presented The diagram represents the whole mariner element using the green bar, at the beginning of which the ARE element is shown. Maybe the ARE element should be presented in the context of the mariner TIRs. 

Paragraph 3.4 and Figure 5. How were the core promoter motif detected? Could you depict all the motif in the context of the TIR?

The authors state that mobilization of the mariner element decrease as the tranascription increases and that this could be due to the intercalating effect of Daunorubicin that may distort the DNA double helix. A simplest explanation could be an inhibitory effect of transposase over expression (also known as OPI, see 10.1093/oxfordjournals.molbev.a025615). Alternatively, it could be due to a lethal effect of the mariner transposition, that affect vital loci. In this context, it could be interesting to estimate the viability of the strain in a genetic background not permissive for the mariner transposition. I would not consider defect in the DNA replication systems since this would be associated to relevant phenotypes, or even to a lethal phenotype.

Author Response

#Rewier 4

This manuscript describe the mobilization of a mariner element inserted in the white gene of D. simulans, upon exposure to chemotherapy drugs, such as Cisplatin, Dacarbazine, and Daunorubicin.

In my opinion the manuscript is well written and the conclusions are supported by the results obtained.

However, I have some major points that I think the authors should clarify. More important, I have found the manual annotation of the ARE elements not appropriate. This analysis should be repeated in a more rigorous way. Finally, some figures should be better presented.

Please, find my specific comments below.

#QUESTION 1 - L36-39 In the context of the functional role that TEs have in the eukaryotic genome, it would be appropriate to cite relevant reviews that have summarized this topic (e.g  10.3390/biology9020025 and 10.1098/rstb.2019.0347) ARE detection. The authors state that the detection of ARE motifs was made manually. They should use more reliable methods such to perform detection, possibly associated to a reliability score or similar. More important, I suggest to check carefully how this analysis was performed. I suspect there is an error in the reported ARE motif (TGACNNNCG) that should be RTGACNNNGC as reported in Itoh et al. 1999, and as also reported in the PWM described in the work of Malhotra et al (10.1093/nar/gkq212). I suggest using tools in RSAT, such as the matrix scan tool (http://rsat.sb-roscoff.fr/matrix-scan_form.cgi), which detect motifs based on PWMs. Nrf2 PWM has been described in the work of Malhotra et al (10.1093/nar/gkq212).

#Answer:  Thank you very much for the suggestions. We have added citations in the introduction and implemented an analysis of the cis-regulatory sites using RSAT software. We have included this new analysis in the text.

#QUESTION 2 - How many ARE motifs can be detected? If many, are they clustered?

#Answer:  We found 3 putative sites, but only one has statistical significance.

#QUESTION 3 - Figure 5. This figure could be better presented. The diagram represents the whole marinerelement using the green bar, at the beginning of which the ARE element is shown. Maybe the ARE element should be presented in the context of the mariner TIRs.

#Answer:   We have modified the figure. 

#QUESTION 4 - Paragraph 3.4 and Figure 5. How were the core promoter motif detected? Could you depict all the motif in the context of the TIR?

#Answer:  We have modified the figure and the text. 

#QUESTION 5 - The authors state that mobilization of the mariner element decrease as the tranascription increases and that this could be due to the intercalating effect of Daunorubicin that may distort the DNA double helix. A simplest explanation could be an inhibitory effect of transposase over expression (also known as OPI, see 10.1093/oxfordjournals.molbev.a025615). Alternatively, it could be due to a lethal effect of the marinertransposition, that affect vital loci. In this context, it could be interesting to estimate the viability of the strain in a genetic background not permissive for the mariner transposition. I would not consider defect in the DNA replication systems since this would be associated to relevant phenotypes, or even to a lethal phenotype.

#Answer:  We agree that overexpression of mariner could be an alternative hypothesis, and we include this possibility in the text. However, cisplatin and daunorubicin have similar increases in transposase transcription, and with cisplatin we observe an increase in red spot. Heat shock also enhances mariner transcription and is followed by transposase excision activity increases (doi:10.1007/s12192-015-0611-2.). Inibition by overexpression, as described by Lohe & Hartl, was not observed in these case (cisplatine and heat shock).

Round 2

Reviewer 1 Report

The authors have clarified experimental issues and improved the text sections.

Author Response

The manuscript was revised by a native speaker.

Reviewer 4 Report

I wish to thank the authors for providing a revised form of the manuscript. All my concerns have been addressed.

I have just some minor changes to suggest.

Figure 1. espectrofluorometric should be spectrophotometric

paragraph 3.4 Please revise the first sentence

Author Response

Figure 1. espectrofluorometric should be spectrophotometric

This error has been corrected in Figure 1 and in the graphical abstract.

Paragraph 3.4 Please revise the first sentence

This paragraph has been rewritten.

The manuscript was revised by a native speaker.